# MicroRNA-29a Mitigates Laminectomy-Induced Spinal Epidural Fibrosis and Gait Dysregulation by Repressing TGF-β1 and IL-6

**DOI:** 10.3390/ijms24119158

**Published:** 2023-05-23

**Authors:** I-Ting Lin, Yu-Han Lin, Wei-Shiung Lian, Feng-Sheng Wang, Re-Wen Wu

**Affiliations:** 1Department of Orthopedic Surgery, Kaohsiung Chang Gung Memorial Hospital, Kaohsiung 83301, Taiwan; jeremy79411@cgmh.org.tw; 2Department of Medicine, Graduate Institute of Clinical Medical Sciences, Chang Gung University College of Medicine, Kaohsiung 83301, Taiwan; cgmhlinyh202@cgmh.org.tw (Y.-H.L.); lianws@cgmh.org.tw (W.-S.L.); wangfs@cgmh.org.tw (F.-S.W.); 3Center for Mitochondrial Research and Medicine, Kaohsiung Chang Gung Memorial Hospital, Kaohsiung 83301, Taiwan; 4Department of Medical Research, Kaohsiung Chang Gung Memorial Hospital, Kaohsiung 83301, Taiwan; 5Core Laboratory for Phenomics & Diagnostics, Kaohsiung Chang Gung Memorial Hospital, Kaohsiung 83301, Taiwan

**Keywords:** laminectomy, spinal epidural fibrosis, microRNA-29a, epigenetic regulation, TGF-β1

## Abstract

Spinal epidural fibrosis is one of the typical features attributable to failed back surgery syndrome, with excessive scar development in the dura and nerve roots. The microRNA-29 family (miR-29s) has been found to act as a fibrogenesis-inhibitory factor that reduces fibrotic matrix overproduction in various tissues. However, the mechanistic basis of miRNA-29a underlying the overabundant fibrotic matrix synthesis in spinal epidural scars post-laminectomy remained elusive. This study revealed that miR-29a attenuated lumbar laminectomy-induced fibrogenic activity, and epidural fibrotic matrix formation was significantly lessened in the transgenic mice (miR-29aTg) as compared with wild-type mice (WT). Moreover, miR-29aTg limits laminectomy-induced damage and has also been demonstrated to detect walking patterns, footprint distribution, and moving activity. Immunohistochemistry staining of epidural tissue showed that miR-29aTg was a remarkably weak signal of IL-6, TGF-β1, and DNA methyltransferase marker, Dnmt3b, compared to the wild-type mice. Taken together, these results have further strengthened the evidence that miR-29a epigenetic regulation reduces fibrotic matrix formation and spinal epidural fibrotic activity in surgery scars to preserve the integrity of the spinal cord core. This study elucidates and highlights the molecular mechanisms that reduce the incidence of spinal epidural fibrosis, eliminating the risk of gait abnormalities and pain associated with laminectomy.

## 1. Introduction

Spinal epidural fibrosis is a well-documented complication that arises after spine surgery, contributing to an alarming 14% of failed back surgery syndrome (FBSS) cases [1]. This condition denotes a significant deterioration in patients’ daily activities while adding an immense burden to healthcare. The etiology of this condition is characterized by overgrown scar tissue, which has been observed to adhere to and encase the dura and nerve roots postoperatively [2]. The affected dura and nerve roots are prone to provoking pain when subjected to back and limb movement traction. Notably, excessive epidural scar tissue has been found to be correlated with the recurrence of radicular pain after lumbar laminectomy and discectomy [3].

Although postoperative scar formation is an inevitability, minimizing epidural fibrosis is a prioritized task [4]. To achieve this aim, several biomaterials and biochemical methods have been found to scale back the formation of epidural fibrosis. The partition between the affected epidural and scar is a common strategy. For example, the partition between the affected epidural and wound with polyethylene film and sterile Vaseline oil has been observed to reduce the occurrence of epidural fibrosis [5]. In experimental animal models, administration of decellularized adipose matrix [6,7], taurine [8], long non-coding ribonucleic acid (lncRNA) [9], hyaluronic acid hydrogel [10], etc. has been found to reduce the extent of epidural fibrotic matrix formation in scar after lumbar laminectomy. These findings suggest that direct and indirect control of scar fibroblasts’ biological function and metabolic activity are alternative strategies for preventing epidural fibrotic scar [11].

Wound repair is mediated by inflammation, angiogenesis, matrix acquisition, and remodeling in the scar. An aberrant homeostasis between fibrosis-promoting and inhibitory pathways is attributable to fibrotic matrix overproduction in tissues [12]. Connective tissue growth factor inhibitor CCN5 signaling reportedly reduces TGF-β1 function and fibrotic matrix collagen and α-smooth muscle actin expression in fibroblasts of scar tissue in rats after laminectomy regulates [13]. All-trans retinoic acid treatment suppresses proinflammatory cytokine IL-6 expression in scars by inactivating the nuclear factor-κB pathway [14,15]. Attenuation of inflammation and fibroblast anabolism appears to delay fibrotic responses within epidural scars via inhibition of TGF-β1/Smad3 and HMGB1/TLR4 signaling pathways [16]. However, the master regulator modulating epidural fibrosis activities remains inconclusive.

MicroRNAs are emerging molecules with small nucleotide sequences that regulate physiological and pathological activities in tissue microenvironments by interrupting the mRNA function of the targets [17]. Of microRNAs, the microRNA-29 (miR-29) family, including miR-29a, 29b, 29c, and 29a-3p, is reported to be a potent anti-fibrosis regulator that participates in the pathogenesis of tissue fibrosis. For example, miR-29b inhibits TGFβ1/Smad signal transduction and decreases dialysis-mediated atrial fibrosis [18]. Loss of miR-29a function elevates fibrotic collagen accumulation, whereas gain of miR-29a action decreases bleomycin-induced lung fibrosis [19]. Promoting miR-29 signaling is found to weaken TGF-β1, fibronectin, and collagen I deposition in dystrophic musculoskeletal tissue [17,20]. Our investigation of transgenic mice overexpressing miR-29a uncovered that miR-29a overexpression enables animals to exhibit minor responses to fibrosis in diabetic kidneys [21,22], ligation-induced liver injury [23,24], and excess glucocorticoid-induced osteoporosis [25,26]. Of interest, treatment with non-coding RNA signaling becomes an innovative therapeutic strategy for excessive fibrosis in various tissues. Nevertheless, the function of miR-29a in spinal epidural fibrosis has not yet been thoroughly studied.

This study aimed to investigate whether miR-29a affected the formation of spinal epidural fibrosis and characterized the epigenetic mechanism by which miR-29a controlled TGF-β1 and IL-6 signaling in the fibrotic activity of laminectomy.

## 2. Results

### 2.1. Mouse Model for Lumbar Laminectomy (LM) Procedures Characterization

In order to examine the potential of miR-29a to inhibit fibrotic activity in scar tissue, we adopted T13-L1 laminectomy (LM) to induce lumbar spinal epidural lesions in a study (Figure 1A). The study involved surgical procedures inducing a spinal cord injury in wild-type (WT) and miR-29a transgenic mice (miR-29aTg). It is crucial to correctly locate the incision site to ensure proper surgical access. First, trace the caudal ribs up to the T13 vertebra. Once the T13 vertebra has been identified, make a precise 1.5–2 mm skin incision along the midline to allow for clear visualization of the lamina at the T12-L2 level. After the incision has been made, it is necessary to carefully dissect the animal’s paravertebral muscles, spinous processes, and lamina at the T13-L1 position. This step is critical to avoiding potential complications and ensuring a successful surgical outcome. It is essential to use sharp dissection techniques to minimize trauma to the surrounding tissues and ensure accurate visualization of the surgical site (Figure 1B).

### 2.2. miR-29a Overexpression Improved LM-Induced Behavior Alteration

The aim is to test whether miR-29a is a plausible candidate for protection against LM-induced persistent injury and fibrosis in the lumbar spinal epidural. Following an experimental surgically induced injury at the desired location, behavioral assessments can be performed to confirm changes in spinal cord tissue integrity and function. Hence, the behavioral examination is one of the basic tests after a therapeutic intervention or comparing beneficial genetic changes among mice, while different behavioral tests estimate various changes in the sensorimotor system of mice [27]. We performed behavioral assays in WT and miR-29aTg mice with or without LM. Monitoring the locomotor behavior in a fixed square arena for 15 min showed that the WT + LM group only moved roughly around the arena while reducing the activity in the middle area (Figure 2A). Statistical behavioral changes in moving time and moving distance were reduced in the WT + LM group than in the miR-29aTg group (Figure 2B). In addition, the spinal epidural injury resulted in irregular hind leg footprints and reduced gait pressure in the WT + LM group (Figure 3A). Significant decreases in hind leg maximum contact strength and footprint area followed in the WT + LM group (Figure 3B). In comparison, miR-29a overexpression improved the footprint histogram and gait contour in lumbar spinal cord injury.

### 2.3. Overexpression of miR-29a Downregulates Fibrotic Matrix Formation and Expression of Related Genes in Lumbar Spinal Cord

Given that the laminectomy-induced disorder of mobile behavior in the WT + LM group is improved by the ubiquitous expression of miR-29a, we further tested whether miR-29a changes fibrotic activity. Histological sections showed extensive scar tissue and fibrotic matrix synthesis in the WT + LM group, but only a small amount of fibrosis in the miR-29aTg + LM group, as indicated by Masson’s trichrome staining (Figure 4A) and collagen fiber quantification (Figure 4B). These pathological features also revealed the expression of fibrotic markers, including immunostaining with TGF-β1 (Figure 5A), IL-6 (Figure 5B), and Dnmt3b (Figure 5C). On the other hand, miR-29aTg significantly attenuated the expression of these markers of fibrotic tissue development in LM surgery (Figure 5D).

## 3. Discussion

Prior clinical observations have noted the importance of the fact that 37% of adults experience lower back pain, and 60–85% of people may encounter this type of pain throughout their lifetime. Additionally, the likelihood of experiencing lower back pain increases as a person ages [28] and can develop as continued lower extremity pain, including leg, back, and lumbar pain. Unexplained persistent low back pain or radicular pain that persists despite following one or more surgical interventions or occurs after surgical intervention is also known as Failed Back Surgery Syndrome (FBSS) [1]. FBSS can be permanent due to various factors, with up to a 20–40% incidence following lumbar surgery that fails to achieve preoperative expectations [29]. Most important is the increased proportion of FBSS patients who develop clumsiness or an inability to walk. Clinical pathological features of FBSS patients have increasing evidence reveals that residual lateral recess/foraminal stenosis, epidural fibrosis, disc degeneration, and herniated nucleus pulposus recurrence [30]. However, molecular mechanisms underlying pathophysiology and etiology identification have not been fully understood and are critical to postoperative care and successful pain management.

Epidural fibrosis, also known as post-laminectomy syndrome, has been recognized as a significant cause of FBSS in patients undergoing laminectomy for reasons such as stenosis or disc herniation. Due to dural entrapment or radix compression, epidural fibrosis may cause radicular symptoms similar to those caused by surgery, reducing the success rate of recurrent surgery to 5–30% [31] and exacerbating symptoms and pain in 10–30% of cases [32]. Repeated surgeries also increase epidural fibrosis rates, creating a vicious cycle of patient distress. Machine learning algorithms evaluate [33] autologous blood and platelet-rich plasma injection [34], high-dose spinal cord stimulation [35], non-steroidal anti-inflammatory drugs (NSAIDs) [36], and epidural adhesiolysis [29] and improve laminectomy-mediated fibrosis progression, inflammation of nerve roots, and pain. A growing body of evidence suggests that microRNA signaling plays a vital role in fibroblast cellular behavior and the development of fibrotic activity originating from surgical scar tissue [37]. In addition, miR-519d-3p [38] and miR-146 [39] may serve as diagnostic biomarkers for the early detection of epidural fibrosis and further upregulation of enhanced fibroblast proliferation and inflammatory response in epidural fibrosis.

Previous studies have demonstrated that miR-29a has multiple biological effects, including attenuation of subacromial bursa fibrosis and shoulder stiffness [40], liver fibrogenesis [41], and it regulates major fibrotic pathways, such as TGF-β1/Smad, PI3K/Akt/mTOR, and DNA methylation [42], concerning the contribution of miR-29a-dedifferentiated formation of fibrosis matrix in spinal epidural scar post-laminectomy. In this study, where laminectomy-mediated fibrotic activity in scar tissue was revealed for the first time, it is worth highlighting the findings on miR-29a and its potential impact on reducing excess production of the fibrotic matrix. miR-29a was indispensable in repressing fibrotic activating factor, TGF-β1, and proinflammatory factor, IL-6, in the scar tissue. This study sheds new light on the microRNA pathway attenuation of fibrosis in the spinal epidural compartment during scar tissue formation with laminectomy. We also convey a microRNA remedial potential for the deterioration of the fibrosis matrix to alleviate spinal epidural dysfunction.

The miR-29 family (a, b, c) of microRNAs is known to control the development of fibrosis in multiple organs. This reaction occurs by pairing its 5’ untranslated region (UTR) with the 3’ UTR of the target mRNA [43]. The miR-29a family has been recognized as a pleiotropic molecule that serves various physiological functions. Our previous research discussed the expression and regulation mechanism of miR-29a in various diseases, such as liver injury/fibrosis [44,45], diabetic nephropathy [21], osteoarthritis [46], and rotator cuff lesion [40], etc., and has proven that miR-29a significantly affects the expression of IL-6, TGF-β, Dnmt3b [47], TNFSF13b [26], FoXO-3 [48], COL3A1 [49], and more [50]. Regardless, ubiquitous expression of miR-29a has systemic regulation and a beneficial impact on various pathological processes throughout the body. A target of the fibrogenesis activator, TGF-β1, is a general cytokine that has been shown to be involved in a wide range of cellular activities and implicated in the pathogenesis of diseases such as connective tissue disease [51], bone disease [48], fibrogenesis [52], and tumorigenesis [53]. TGF-β1 plays a crucial role in initiating and progressing tissue fibrosis; its effector Smad proteins (Smad2, Smad3, and Smad4) have multiple bidirectional roles in regulating fibrosis. Smad3 has the property of directly binding to the Smad binding site to promote transcription, while Smad2 and Smad4 assist Smad3 in regulating gene transcription [54]. TGF-β1 was associated with Smad proteins to increase the transcriptomic activity of IL-6 [55]. In addition, TGF-β1 also upregulates the transcription activity of the DNA methyltransferase factor, Dnmt3, to promote collagen secretion [56]. The miR-29 family has been reported to negatively regulate TGF-β1 expression by directly targeting the 3’UTR region of SERPINH1 (encoding the heat shock protein HSP47) [51] and can also affect endometrial fibers TGF-β1/Smad signaling pathway [57]. Also, miR-29a inhibits the TGF-β2/Smad3 signaling pathway, thereby reducing the formation of skin scar tissue [58]. Consistent with the miR-29a transgenic mice (miR-29aTg) analysis, mild fibrotic activity and TGF-β1 expression were demonstrated in laminectomy-mediated spinal epidural fibrosis.

This study demonstrated that in a spinal epidural injury model, miR-29aTg exhibited favorable responses to irregular walking patterns and gait disturbances induced by laminectomy-mediated epidural fibrosis. Immunohistology analysis also demonstrated the downregulation of Dnmt3b and IL-6 expression in spinal epidural scars. miR-29a can directly target the 3’-UTR of Dnmt3b to reduce DNA methylation modification in developing nonalcoholic steatohepatitis [47]. Nevertheless, increasing evidence exists that disruption of Dnmt3b signaling components attenuates tissue pathogenesis associated with delayed fibrosis, inflammation, and pain [56,59]. Understanding the action of epigenetic modification and the crosstalk between miRNAs and Dnmts involved in fibrogenic activity is beneficial for treatment strategy development [55,60]. Notably, miR-29a precursor treatment has been shown to compromise fibrosis in shoulder stiffness [40] and fibrogenesis of keloid [61] in animal studies. This study highlights the salutary effect of miR-29a precursor treatment in delaying pathological fibrotic remodeling.

In conclusion, we do not exclude the possibility that miR-29a targets other fibrogenic factors or extracellular matrix pathways during chronic fibrosis and acknowledge the limitation of this study in that the anatomy or biomechanics of spinal epidural injury and scar tissue in mice cannot be fully extrapolated to human FBSS. FBSS refers to a subset of patients who undergo new or persistent pain after spinal surgery for back or leg pain, while epidural fibrosis is a common cause of FBSS. Regardless of the patient’s condition, treatment should be multidisciplinary and individualized. In this sense, clinical treatment includes surgical and non-surgical treatment, but its efficacy is still controversial [62,63]. Previous articles showed that Chinese herbal medicines, including berberine and evodiamine, are associated with increased endogenous miR-29a and its target gene DNMT3A/3B [64]. Nonetheless, using the lentivirus vector encoding the miR-29a precursor to treat fibrotic tissue has been highly effective in animal studies. However, safety concerns have led to a lack of intensive remedies in clinical practice. Evidence from a growing body of research suggests that specific cell-derived extracellular vesicles promote endogenous or circulating miR-29a expression and may encourage the development of future clinical therapeutic strategies [65]. This study presented microRNA insight into miR-29a, a critical regulator of fibro-inflammatory processes in laminectomy-mediated spinal epidural fibrosis. It may protect the body from fibrosis by regulating epigenetic factors, and further investigations are needed to explore the potential of miR-29a in treating fibrosis related to FBSS.

## 4. Materials and Methods

### 4.1. Transgenic Mice Overexpressing MiR-29a

The protocols for breeding, experimenting, and caring for the animals were reviewed and approved by the Institutional Animal Use and Care Committee, Kaohsiung Institutional Animal Use and Care Committee (Affidavit No. 2016122603). FVB mice were used to overexpress the miR-29a precursor (FVB/TNar-Tg-29a/PGK; miR-29aTg) driven by the PGK (phosphoglycerate kinase) promoter, as previously described [26,43]. The genotype of each animal was confirmed using customized primers (forward: 5′-GAGGATCCCCTCAAGGATACCAAGGGATGAAT-3′; reverse, 5′-CTTCTAGAAGGAGTGTTTCTAGGTATCCGTCA-3′) for PCR analysis.

### 4.2. Lumbar Laminectomy Model

The animal surgery procedures and the protocol under study are briefly described below: The mouse was placed in a prone position with its extremities immobilized under general anesthesia and sterile conditions. A 3-cm longitudinal skin incision was made along the vertebral column on the mid-back. Soft tissues and paraspinal muscles were dissected, and the spinal process and lamina were carefully exposed. The lumbar spine was subjected to a two-stage laminectomy that included the removal of the spinous processes, lamina, and ligamentum flavum. An operating microscope carefully exposes the underlying dura (Carl Zeiss, Inc., Jena, Germany). Hemostasis was achieved, and the wound was irrigated with saline. 3-0 Vicryl suture was used to close the fascia and skin [66].

### 4.3. Assessment of Gait Profile and Mobility Patterns

Animal footprint characteristics were assessed using the Noldus Catwalk gait analysis system (Noldus Information Technology, XT). After 8 weeks of laminectomy and sham operation, the gait pattern was observed and recorded using a high-speed video camera and the Catwalk software (V. 9.1, Leesburg, VA, USA) of the device while the animal is walking on a 100 cm gangway. Each affected hindlimb’s footprint and gait profile were calculated, including footprint area, swing time, speed, and pressure. In addition, animal behavioral activities were assessed using an open field arena measuring 30 × 30 × 30 cm (w × d × h). The mice were positioned in the center of the arena. After 5 min of habituation, the system automatically followed and recorded animal movement patterns and activities for 15 min (Multi Conditioning System, TSE Systems GmbH, Bad-Homburg, Germany).

### 4.4. Lumbar Tissue for Fibrotic Activity Quantification and Immunohistochemical Analysis

Mouse lumbar spine tissues were fixed in 10% paraformaldehyde for 48 h, then immersed in decalcification buffer (Epredia™, Fisher Scientific, Waltham, MA, USA) until a 27-gauge needle could easily pass through the tissue, and the tissues were embedded in paraffin. Tissue slides were deparaffinized, and continuous 5-mm sections were subjected to Masson’s trichrome staining (Polysciences, Bayonne, NY, USA) according to the manufacturer’s standard protocol. For immunohistochemistry, tissue sections were deparaffinized and subjected to epitope retrieval procedures at 95 °C for 30 min (Antigen Retrieval Reagent, Enzo Life Science, Farmingdale, NY, USA). Primary antibodies against IL-6 (ab9324), TGF-β1 (ab215715), and Dnmt3b (ab2851, Abcam, Boston, MA, USA) were detected in sections using BioGenex detection kits (BioGenex, Fremont, CA, USA). Histological images were captured using a digital slide scanner (Pannoramic MIDI, 3DHISTECH Ltd., Budapest, Hungary), and images were randomly selected for quantification under constant magnification using ImageJ (MacOS 10.12 (Sierra)) from five fields of each section [67].

### 4.5. Statistical Analysis

Data were expressed as means ± SEM. Differences among miR-29a transgenic mice and wild-type mice with laminectomy or sham operation. A parametric ANOVA test and a Tukey post-hoc test analyzed groups between comparative. For all analyses, *p* < 0.05 was considered statistically significant and performed using the Statistical Package for GraphPad Prism 9 (GraphPad Software, San Diego, CA, USA).

## Figures and Tables

**Figure 1 ijms-24-09158-f001:**
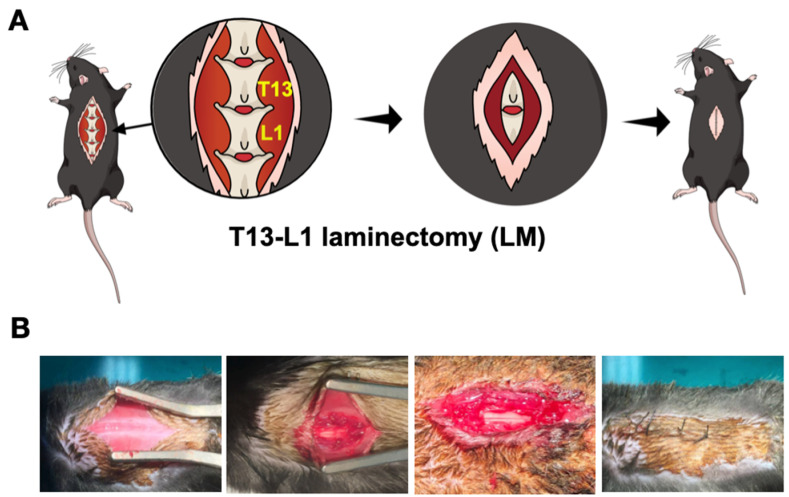
Schematic diagrams showing laminectomy (LM) to expose the lumbar spinal cord of mice. Cartoon schematic diagram showing surgery-relevant locations (**A**), and exposed lumbar spinal cord area for laminectomy procedures (**B**).

**Figure 2 ijms-24-09158-f002:**
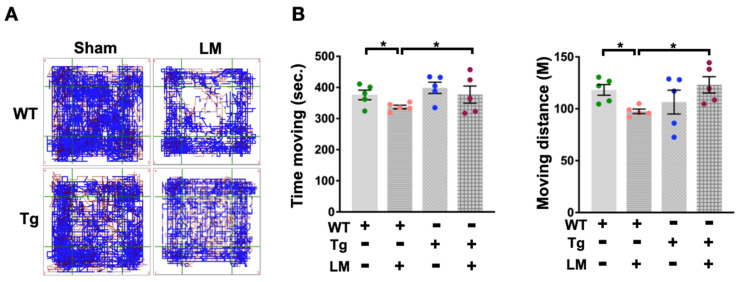
Assessment of locomotor behavior in LM mice. Individual observation trajectories showed changes in exploratory activity in WT and miR-29aTg mice. Red stripes signify locomotion (horizontal movements), and blue stripes signify rears (vertical movements) in a square arena. (**A**) Exploratory activity was calculated as the moving time and distance (**B**). Dots represent the number of analyze in each group, while colors define different groups. Data are shown as mean ± SEM and analyzed by ANOVA test followed by Tukey post-hoc test. * *p* < 0.05 indicate significant differences between groups.

**Figure 3 ijms-24-09158-f003:**
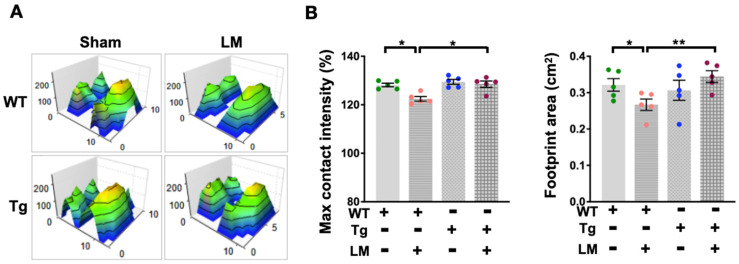
Monitor and quantify gain profiles in LM mice. Respective exploration LM-mediated irregular 3D footprint pressure patterns (**A**), In miR-29aTg mice, changes in calculated maximum contact strength and footprint area were ameliorated in lumbar spinal cord injury (**B**). Dots represent the number of analyze in each group, while colors define different groups. Data are shown as mean ± SEM and analyzed by ANOVA test followed by Tukey post-hoc test. * *p* < 0.05., ** *p* < 0.01 indicate significant differences between groups.

**Figure 4 ijms-24-09158-f004:**
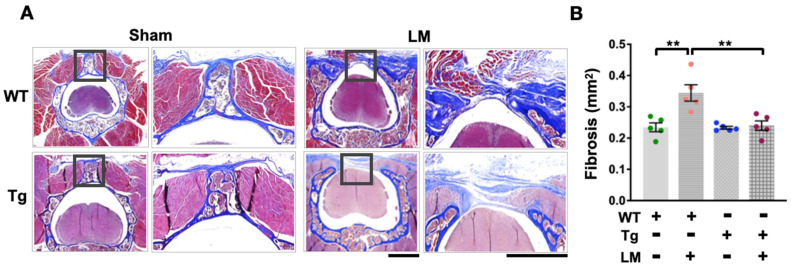
Analysis of fibrosis histopathology and fibrotic matrix expression of the lumbar spinal cord in WT and miR-29aTg mice with or without LM. Masson’s trichrome staining displayed spacious fibrotic tissue (blue) in the WT + LM group (**A**). Exploratory fibrotic matrix activity was quantified (**B**). Dots represent the number of analyze in each group, while colors define different groups. Data are shown as mean ± SEM and analyzed by ANOVA test followed by Tukey post-hoc test. ** *p* < 0.01 indicate significant differences between groups.

**Figure 5 ijms-24-09158-f005:**
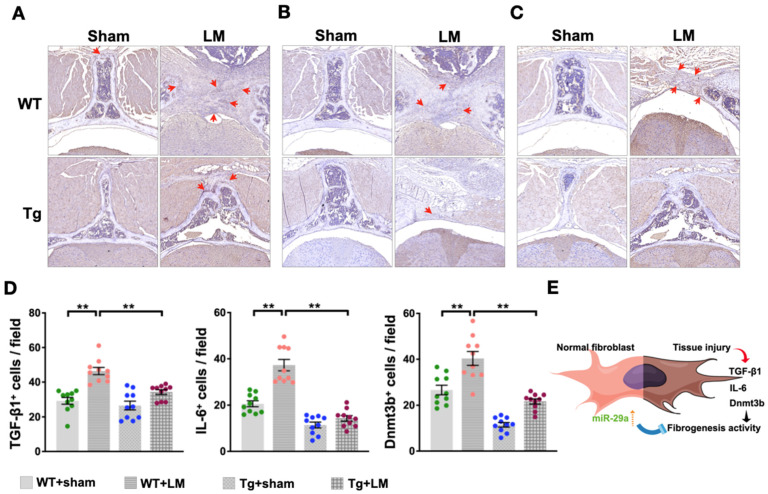
Histological analysis of LM-induced lumbar spinal cord injury. Lesion sites in the miR-29aTg group showed slight immunoreactivity in TGF-β1 (**A**), IL-6 (**B**), and Dnmt3b (**C**), expression signaling as indicated by the red arrows. Furthermore, overexpression of miR-29a significantly decreased TGF-β1, IL-6, and Dnmt3b signaling (**D**). Schematic graphs showing miR-29a protection against tissue injury development fibrotic activity. The miR- 29a-mediated downregulation of the expression of TGF-β1, IL-6, and Dnmt3b in fibrogenesis matrix (**E**). Dots represent the number of analyze in each group, while colors define different groups. Data are shown as mean ± SEM and analyzed by ANOVA test followed by Tukey post-hoc test. ** *p* < 0.01 indicate significant differences between groups.

## Data Availability

The data supporting this study’s findings are available on request from the corresponding author.

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
