# Peer review of "MicroRNA-29a Mitigates Laminectomy-Induced Spinal Epidural Fibrosis and Gait Dysregulation by Repressing TGF-β1 and IL-6"

_ijms, 2023, doi:10.3390/ijms24119158_

Round 1

Reviewer 1 Report

Major comments:

1. This study used transgenic mice that ubiquitously express miR-29a to study its effect on the LM-induced spinal epidural fibrosis. Is there a characterization of this animal model to evaluate the global gene expression changes in comparison with the WT? i.e. Does the ubiquitous miR-29a expression introduce a phenotype before the LM surgery?

2. In Figure 5, the authors quantified the TGF-b1, IL-6 and Dnmt3b positive cells/field. Is it possible to perform more quantitative methods to consolidate the conclusions, such as qRT-PCR and Western-blot?

3. miR-29a seems to be a promising new treatment for LM-induced spinal epidural fibrosis based on the results from this manuscript. Could the authors address how to deliver the microRNAs to patients after LM surgery? How to determine the dosage for the treatment?

Minor comments:

1. It is suggested to use patterns in bar graphs to help colorblind readers to visualize the data.

2. The down-regulation of Dnmt3b is an indirect evidence of reduced DNA methylation. Direct assays to detect DNA methylation may be needed to draw this conclusion.

Minor editing of English language is recommended.

Reviewer 2 Report

The manuscript “MicroRNA-29a Mitigates Laminectomy-Induced Spinal Epidural Fibrosis and Gait Dysregulation by Repressing TGF-β1 and IL-6” by  I-Ting Lin et al. aimed to investigate whether miR-29a affected the formation of spinal epidural fibrosis and characterized the epigenetic mechanism by which miR-29a controlled TGF-β1 and IL-6 signaling in the fibrotic activity of laminectomy.

Below are my comments and remarks regarding the manuscript:

1. epidural panic what is it?

2. T13-T1 please correct to T13-L1

3. methodology

what did the author mean by spinal cord injury?

  it should be spinal epidural injury,

during laminectomy we avoid damage to nerve structures. Fibrosis itself does not cause damage to the spinal cord, or only indirectly. Was the spinal cord or dura  exposed in the spinal canal? This should be clarified

4. injury progressive deterioration of the lumbar spinal. This sentence needs improvement.

5. How long after the operation were behavioral tests performed?

6. Discussion

what does mean HBSS ?

7. Post-laminectomy membrane - it should be post-laminectomy syndrome

8. discussion

A schematic figure of the pathways responsible for progressing tissue fibrosis would be useful

Round 2

Reviewer 1 Report

The authors' responses are acceptable. Please incorporate them (especially to major comment 1 and 3) into the main text if possible.

Author Response

We are grateful to the reviewers for acknowledging our study. In the following, our changes to the manuscript are highlighted in red.

1. A reply to Major comment 1 has been integrated into the Discussion section, lines 207-214.

2. A reply to Major comment 3 has been integrated into the Discussion section, lines 247-259.

Reviewer 2 Report

I have no more comments

Author Response

The research team like to thank you for your precious time and advice.